# Serial 'deep-sampling' PCR of fragmented DNA reveals the wide range of *Trypanosoma cruzi* burden among chronically infected human, macaque, and canine hosts, and allows accurate monitoring of parasite load following treatment

Brooke E White[1], Carolyn L Hodo[2], Sarah Hamer[3], Ashley B Saunders[4], Susana A Laucella[5], Daniel B Hall[6], Rick L Tarleton[1,7]*

[1]Center for Tropical and Emerging Global Disease, Athens, United States; [2]Michale E. Keeling Center for Comparative Medicine and Research, The University of Texas MD Anderson Cancer Center, Bastrop, United States; [3]Department of Veterinary Integrative Biosciences, School of Veterinary Medicine and Biomedical Sciences, Texas A&M University, College Station, United States; [4]Department of Small Animal Clinical Sciences, College of Veterinary Medicine and Biomedical Sciences, Texas A&M University, College Station, United States; [5]Research Department, Instituto Nacional de Parasitología "Dr. Mario Fatala Chaben", Buenos Aires, Argentina. Chagas Disease Unit, Hospital Interzonal General de Agudos Eva Perón, Buenos Aires, Argentina; [6]Department of Statistics, University of Georgia, Athens, United States; [7]Department of Cellular Biology, University of Georgia, Athens, United States

*For correspondence:
tarleton@uga.edu

Competing interest: The authors declare that no competing interests exist.

## eLife Assessment

This study presents an **important** methodological advance to improve the sensitivity of PCR for detecting Trypanosoma cruzi in blood, combining DNA fragmentation, deep sampling, and blood cell pellet analysis. The findings offer **solid** evidence of enhanced detection sensitivity and shed light on parasite load dynamics during chronic infection in mammalian reservoirs. The evidence is sound for macaques and the method shows promise in expanding detection limits, but there is some variability in the limits of detection and small sample size of human samples. This work will be of interest to parasitologists, epidemiologists, and clinicians using molecular diagnostics to monitor responses to etiological treatments for Chagas disease.

**Abstract** Infection with the protozoan parasite *Trypanosoma cruzi* is generally well-controlled by host immune responses, but appears to be rarely eliminated. The resulting persistent, low-level infection results in cumulative tissue damage with the greatest impact generally in the heart in the form of chagasic cardiomyopathy. The relative success in immune control of *T. cruzi* infection usually averts acute phase death but has the negative consequence that the low-level presence of *T. cruzi* in hosts is challenging to detect unequivocally. Thus, it is difficult to identify those who are actively

infected and, as well, problematic to gauge the impact of treatment, particularly in the evaluation of the relative efficacy of new drugs. In this study, we employ DNA fragmentation and high numbers of replicate PCR reaction ('deep-sampling') and to extend the quantitative range of detecting *T. cruzi* in blood by at least three orders of magnitude relative to current protocols. When combined with sampling blood at multiple time points, deep sampling of fragmented DNA allowed for detection of *T. cruzi* in all infected hosts in multiple host species, including humans, macaques, and dogs. In addition, we provide evidence for a number of characteristics not previously rigorously quantified in the population of hosts with naturally acquired *T. cruzi* infection, including, a >6 log variation between chronically infected individuals in the stable parasite levels, a continuing decline in parasite load during the second and third years of infection in some hosts, and the potential for parasite load to change dramatically when health conditions change. Although requiring strict adherence to contamination–prevention protocols and significant resources, deep-sampling PCR provides an important new tool for assessing therapies and for addressing long-standing questions in *T. cruzi* infection and Chagas disease.

## Introduction

Chagas disease, the result of infection with the protozoan *Trypanosoma cruzi*, is endemic to the Americas, where it is among the highest impact infectious diseases, and is also a major source of infection-related heart disease globally. Chagas disease is a result of the long-term persistence of *T. cruzi* primarily in muscle tissues, despite highly effective immune responses that generally control but fail to completely clear the infection in the majority of individuals. Although *T. cruzi* continuously alternates between replicating forms inside host cells and non-replicating forms in extracellular spaces, including the bloodstream where it can be acquired by blood-feeding triatomine insects, detection of parasites or parasite products in the blood is generally undependable, using even the most sensitive methods. Consequently, diagnosis of infection generally rests mainly on serological tests, which are often not fully reliable.

Positive serological tests reflect prior exposure but not necessarily active infection. Thus, determining the effectiveness of current treatments to clear the infection or whether some subjects have spontaneously resolved the infection (apparently rare, but anecdotally reported) and thus should not be treated, remains out of reach. The inability to routinely and sensitively detect active infection coupled with the undependable curative capabilities of current therapeutics and their substantial side effects, accounts for the estimates that less than 1% of *T. cruzi*-infected individuals receive anti-parasitic treatment (*Cucunubá et al., 2017*). Furthermore, the lack of sensitive methods to definitively establish cure hinders the identification and validation of improved therapies.

Amplification techniques such as PCR can specifically enrich very low quantities of pathogen DNA and have been extensively used to enhance the detection of *T. cruzi* DNA in the blood of infected hosts. Multiple, high copy number (>$10^5$ copies per organism) targets for amplification have been identified and rigorous, highly specific amplification protocols have been developed and evaluated (*Schijman et al., 2011*). Nevertheless, it is generally agreed that these protocols, as currently employed, frequently fail to detect *T. cruzi* in infected hosts and thus are not reliable tests of the absence of infection. Multiple studies have attempted to address the sensitivity and specificity of the *T. cruzi* PCR methodology, for example, varying the target (there are two primary ones used, a kDNA and a genomic satellite sequence), primer and probe sequences, DNA storage and purification techniques (automated or not), and volume of blood drawn for DNA isolation (1–10 ml) (*Sulleiro et al., 2019*; *Silgado et al., 2020*). In general, none of these variations significantly alter test outcomes. What does alter the ability of PCR to detect *T. cruzi* infection is the number of independent blood collections done and the number of PCR determinations conducted for each sample (*Parrado et al., 2019*).

These results highlight that detection of *T. cruzi* DNA in infected hosts is frequently a problem of sampling. Specifically, when pathogen numbers are low, collection of a high number of test samples that are extensively subsampled may be required in order to have an opportunity to detect a pathogen. For *T. cruzi*, this sampling problem is strongly supported by the remarkable study done in the 1970s by Cerisola in Argentina (*Cerisola, 1977*) in which 30 untreated subjects submitted to approximately monthly xenodiagnosis with 80 triatomines each month for >2 years. This study showed that some individuals had multiple bugs positive at every sample point (thus, presumably a high parasite load)

**eLife digest** Chagas disease is a dangerous tropical illness caused by single-cell parasites known as *Trypanosoma cruzi*. In most cases, if not treated immediately, the infection becomes chronic: the immune system of the host greatly reduces the number of parasites present in the body yet fails to fully eradicate them. Current diagnostic approaches often fail to detect these low numbers of parasites. More broadly, without a reliable way to measure *Trypanosoma cruzi* levels, researchers and clinicians struggle to test new treatments as well as determine whether patients carrying more parasites tend to develop more severe disease.

In response, White et al. aimed to develop a new way of measuring parasitic loads. They based their approach on standard PCR (polymerase chain reaction), an experimental method that amplifies specific DNA sequences in proportion to their initial levels in a sample. This allows researchers to not only pinpoint the presence of the parasites, but also to assess their relative number However, the approach has limited sensitivity: if the amount of target DNA initially collected is too low, it may not be detected.

To bypass this limitation, White et al. adopted a 'deep-sampling PCR' approach and made several crucial changes. First, they collected several samples from the same patient, therefore increasing the overall sampling volume. Second, they fragmented the sample DNA before the amplification step in order to disperse the target DNA, thus increasing the chances of its detection in each PCR reaction. Third, they performed as many as 400 PCRs per sample. These modifications greatly improved sensitivity compared to usual approaches.

White et al. used their newly improved method to examine *Trypanosomas cruzi* levels in infected macaques over long periods. The results show that parasitic burden can be stable over a year, but vastly differs between individuals (with some having a million times more parasites than others). Similar findings were also observed in humans and dogs.

The method developed by White et al. is likely to be too labour intensive to be routinely used in diagnostic laboratories; however, it represents an important and immediate advance for researchers testing new compounds to treat Chagas disease.

while others were more variable (strongly positive some months but not others), and a few were only very occasionally positive (in the most extreme case, as few as 2 positive bugs out of >1000 fed over 24 months).

In the current study, we have used humans, non-human primates (NHPs), and dogs, all with naturally acquired *T. cruzi* infections, to demonstrate that serial sampling of blood and exhaustive PCR of optimally prepared DNA from that blood, can confirm even the most difficult to detect infections with *T. cruzi*. The ability to obtain relative quantification of parasite load in these naturally infected hosts reveals the wide range of parasite load among infected subjects in a population (>6 $\log_{10}$) and the conditions under which parasite control can change, both slowly in the early years of infection and dramatically when overall health changes. Deep-sampling PCR, while laborious, offers the first true test of cure for Chagas disease and should also assist in the determination of associations between parasite load, immune response parameters, and the risk for disease development in chronically infected hosts.

## Results

### Performing large numbers of replicate PCR reactions ('deep-sampling') extends the range to detection of *T. cruzi* infection in naturally infected hosts

In order to determine the potential of deep-sampling for more sensitive detection of *T. cruzi* in blood samples, we spiked macaque blood with DNA equivalent to known numbers of parasites and conducted 40 replicate PCR reactions per DNA sample. A blood DNA aliquot for PCR amplification equivalent to $10^{-3}$ parasites per reaction was selected as the highest parasite concentration as this is the lowest standard that consistently provides a positive signal in our standard PCR assay. This result was confirmed in this assay when all 40 aliquots expected to contain $10^{-3}$ parasites were positive for

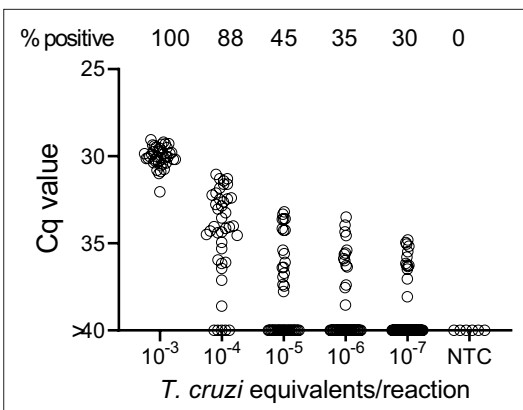

**Figure 1.** Deep-sampling (replicate PCR) allows detection of *T. cruzi* in a decreasing frequency of replicate reactions to at least four orders of magnitude below the normal limit of quantitation ($10^{-3}$) used for single PCR reactions. % positive is the percentage of replicate reactions that gave a detectable amplification in 40 cycles (Cq value <40). NTC = no *T. cruzi* DNA.

The online version of this article includes the following source data for figure 1:

**Source data 1.** Source data for **Figure 1**.

*T. cruzi* DNA amplification (**Figure 1**). Decreasing the parasite equivalents (PE) in 10-fold increments expectedly reduced the frequency of positive PCR reactions and as well, the maximum Cq value of each positive amplification set, but demonstrated that as few as $10^{-7}$ PE per reaction could be detected if a sufficient number of replicate PCR reactions were conducted.

We next turned to the use of *bona fide* blood samples from *T. cruzi*-infected hosts to test the ability of deep-sampling PCR to detect *T. cruzi* DNA in naturally infected hosts with an expected variability of circulating parasites. NHPs in indoor/outdoor housing in the southern U.S. readily acquire *T. cruzi* by environmental exposure to the often plentiful infected triatomine insects present in these settings (**Hodo et al., 2018**) and maintain immunologically controlled but persistent infections similar to that in humans (**Padilla et al., 2022**; **Padilla et al., 2021**).

In total, 21 *T. cruzi* seropositive, two seronegative, and five previously treated and cured (**Padilla et al., 2022**) rhesus macaques were used in the study. The average age of the seropositive macaques was 13.4 years (range 8–24) and the presumed period of infection (based upon first positive serology in annual sampling) was between 2 and 11 years (mean = 5.7) (full data on all the animals is provided in **Supplementary file 1**).

Samples were collected and processed as shown in **Figure 2A**. In brief, three approximately 5 ml blood samples were collected from the same needle stick and DNA was isolated from two of the whole blood samples and the third sample was submitted to hemoculture. PCR amplification of

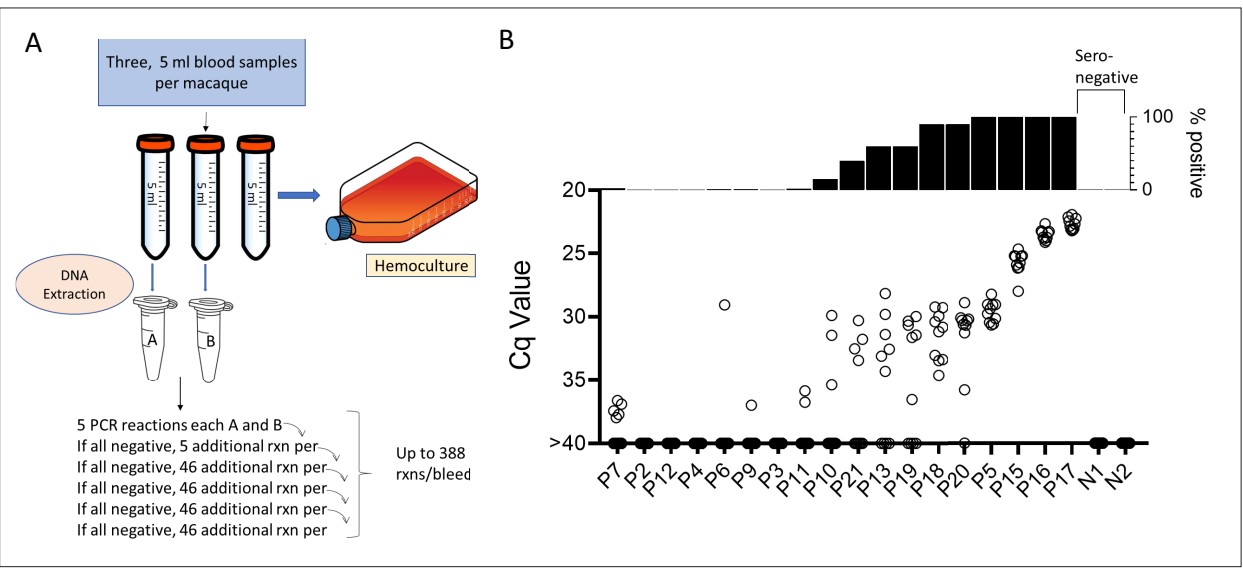

**Figure 2.** Monthly screening protocol for macaques. (**A**) Protocol for the collection and PCR analysis of non-human primate (NHP) blood samples. (**B**) The combined PCR results of samples A and B at the first of monthly sampling points, plotting Cq values for all replicates (bottom) and percent positive replicates (top).

The online version of this article includes the following source data for figure 2:

**Source data 1.** Source data for **Figure 2**.

replicate 125 ng samples of the blood DNA was conducted for each of samples A and B as noted until a positive reaction was observed in either sample or until the DNA was depleted, up to a total of 194 amplification reactions per sample. For the 18 seropositive macaques included in the initial sampling point, 11 (61%) had at least 1 positive amplification in 10 or fewer PCR reactions and 6 of these 11 were positive on 9 or 10 of the 10 reactions and thus likely to be detected by a standard PCR (one or two replicate) test (*Figure 2B*). On the other extreme, 4 of 18 (22%) seropositive macaques were PCR negative despite as many as 388 replicate reactions. Likewise, blood DNA from seronegative controls was negative in >300 replicate reactions (*Supplementary file 1*).

To determine the constancy of detectable parasite DNA in blood over time, blood samples were collected monthly for up to 1 year and processed as in *Figure 2A*. Profiles of representative animals over this year of sampling are shown in *Figure 3*; the full dataset for all 28 animals involved in the study is shown in *Figure 3—figure supplement 1*. All 21 seropositive macaques had one or more positive PCR reaction at one or more of the sample points over the initial 12-month study period. *Figure 3A–C* shows a representative set of animals in which PCR reactions were, respectively, very rare, variable or frequent. Macaques P7 and P9 were two of the most extreme cases in terms of a low frequency of positive PCR reactions, with, respectively, 7 (0.21%) and 10 (0.29%) positive reactions from a total of >3000 reactions performed on 24 blood samples each. The example macaques shown in *Figure 3B* had higher frequencies of PCR positive reactions (1.76, 2.09, and 4.77) and with lower Cq values, but also had occasional months with no positive detection. Collectively 8 of the 21 seropositive macaques had 1 or more months in which as many as 388 replicate PCR reactions failed to detect an amplifiable product. Pearson correlation analysis revealed a very strong negative correlation between the overall frequency of positive PCR reactions in animals and the average Cq values for those positive reactions (low Cq values represent higher target DNA; *Figure 3E*). All macaques screened for a minimum of 12 months in the first year of the study or the eight additional bleed points in the second year (see below) were positive by hemoculture in one or more samples with one exception and the percent positive hemoculture correlated strongly with both the frequency of positive PCR reactions and the average Cq values of positive PCRs (*Figure 3E, F*). Thus, assessing the frequency of positive PCR reactions among deep-sampled, serially collected blood-derived DNA blood allows highly sensitive and quantitatively accurate determination of infection status and relative parasite abundance in naturally infected macaques with a wide range of parasite burdens.

## Serial deep-sampling PCR can reveal changes in parasite burden over time

In general, the frequency of positive PCRs and the Cq values of those positive reactions were relatively stable in macaques across the 1-year study period. The major exceptions to this rule are shown in *Figure 3C, D*. The three macaques in *Figure 3C* had among the highest fraction of positive PCR reactions (70, 81, and 68 %) in the study, but also displayed an increase in mean Cq values over the 1-year study period. Because of the consistent detection of positive PCR reactions (minimum 1 positive reaction in a maximum of 20 replicates every month) in 9 of the 18 initially sampled seropositive macaques (*Figure 1*), these macaques were only sampled for the first 6 months of the study period. However, for these macaques in *Figure 3C* with a pattern of increasing Cq values over 6 months, an additional set of samples was collected at month 12. In all three cases, a highly significant (p < 0.001) trend of increasing Cq values over time was evident, indicating a decreasing parasite load over the 1-year period. Not surprisingly, these three macaques were also among the most recently infected macaques studied (minimum 2–4 years post-infection) and likewise, the macaques in *Figure 3A* with a low frequency of PCR positive reactions were among the longest infected of the macaques under study (6–8 years). However, several macaques did not match this trend of lower parasite load with increasing length of infection (e.g. P11, infected for 2 years, had a low parasite load and P5 and P1, infected, respectively, for 8 and 10 years, had among the highest frequency of PCR positive reactions). There was no statistically significant correlation between the length of infection and the frequency of PCR positive reactions (p = 0.081) or the mean Cq value of positive PCR reactions (p = 0.11) among all macaques in the study (*Figure 3E*). This monthly monitoring of blood parasite DNA thus reveals a relatively stable parasite load over 12 months but huge differences when compared to the spiked samples (*Figure 1*) in the load between individual naturally infected macaques, irrespective of the length of infection.

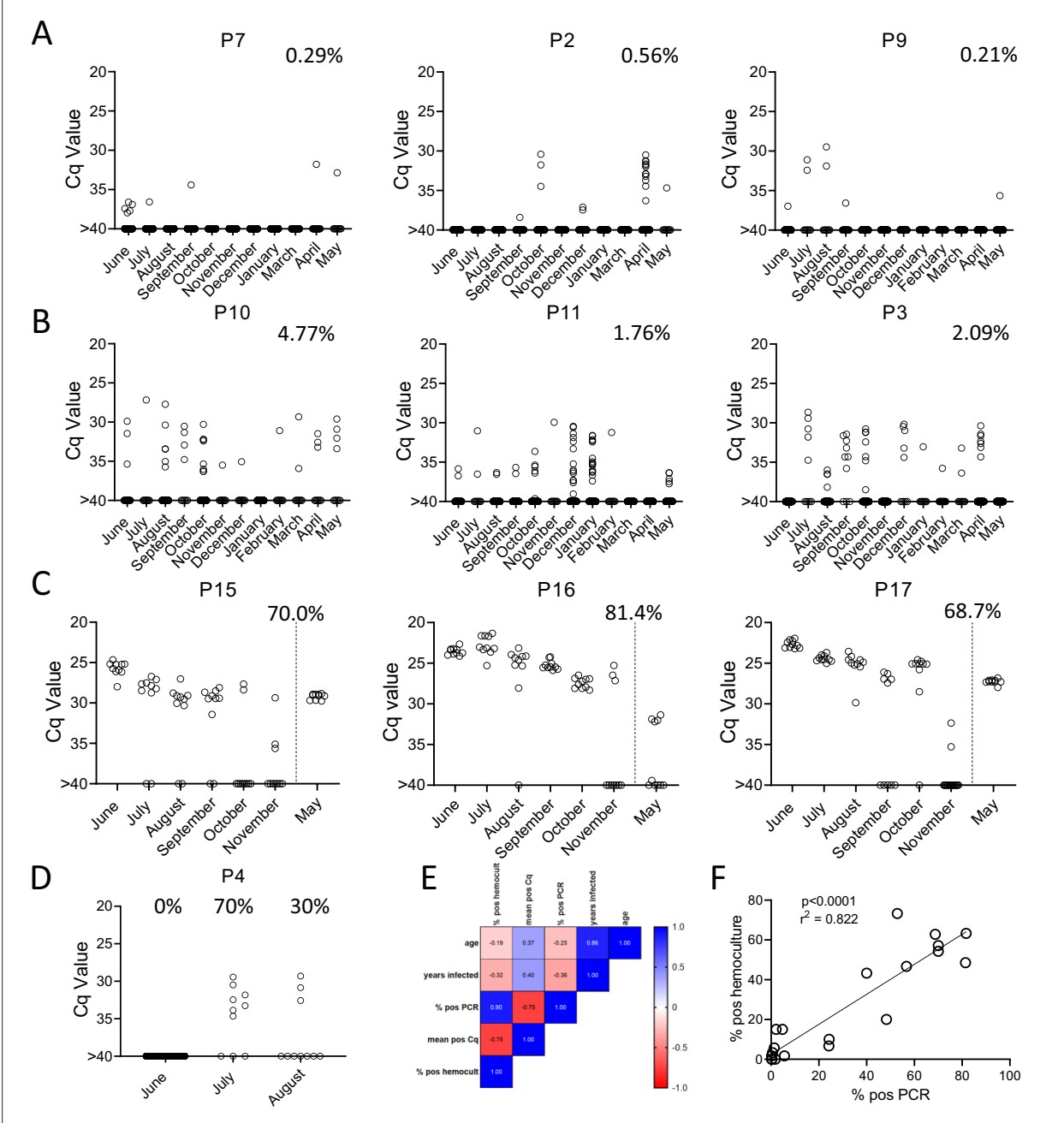

**Figure 3.** Monthly tally of representative macaques with (**A**) rarely, (**B**) variably, or (**C**) frequently positive PCR reactions over the 1 year of sampling. Animals in groups (**A**) and (**B**) were sampled all 12 months while those in (**C**) were sampled only for the first 6 months and then in the 12th month. Percentages indicate the overall percentage of positive PCR reactions across all sampling points. (**D**) Macaque P4 switched from 100% negative to 50% positive PCR reactions coincident with a change in health status. (**E, F**) Pearson correlation analysis indicated a strong positive correlation between the overall frequencies of positive PCRs and hemocultures and a negative correlation between these frequencies and Cq values, but no significant correlation between age or length of infection with any of the three parasite parameters.

The online version of this article includes the following source data and figure supplement(s) for figure 3:

**Source data 1.** Source data for *Figure 3*.

**Figure supplement 1.** The monthly pattern of detection of *T. cruzi* in replicate PCR reactions in DNA from macaque blood collected over 1 year of sampling.

**Figure supplement 1—source data 1.** Source data for *Figure 3—figure supplement 1*.

The other exception to a relatively stable level of *T. cruzi* DNA in blood of infected macaques is shown in *Figure 3D*. Macaque P4 was PCR negative on 384 replicate reactions in the first month of sampling, but became strongly positive in a combined 50% of the 20 replicate reactions in months 2 and 3 of the study, indicating a rapid and dramatic change in *T. cruzi* DNA in blood. This animal developed uncontrolled diarrhea late in the third trimester of pregnancy and had to be euthanized shortly after collection of the August sample. Necropsy and histopathology revealed chronic typhlocolitis with locally extensive ulceration at the ileocecal junction suggestive of a pre-neoplastic or early neoplastic lesion, as well as mild cardiac fibrosis with minimal lymphohistiocytic infiltrate. The origins of the loss of immune control of *T. cruzi* infection in this animal were not further explored, but this episode emphasizes that a well-contained chronic *T. cruzi* infection can rapidly yield to a high-level/uncontrolled infection.

Lastly, with the exception of a set of samples that were cross-contaminated during DNA isolation in the February batch (see Materials and methods), and a single replicate PCR for N1, the two seronegative macaques as well as five macaques previously infected but cured by treatment with benzoxaborole AN15368 (*Padilla et al., 2022*) were PCR negative in a collective 15,084 PCR reactions on samples obtained at 6–11 sampling points (*Figure 3—figure supplement 1*) and were likewise hemoculture negative in all samples.

## DNA fragmentation further increases the sensitivity of detection of *T. cruzi* DNA in blood

We noted that with the input of $10^{-3}$ PE per replicate PCR reaction, which consistently gives 100% positive reactions (*Figure 1*), that these positive reactions were mostly clustered at a relatively low Cq value. However, decreasing the amount of spiked-in parasite DNA to $10^{-4}$ PE or lower resulted in a wide spread in the Cq values, in addition to increasing frequency of negative reactions with decreasing PE/reaction. The parasite DNA sequence targeted for amplification in these assays is a 'satellite DNA' sequence that is represented in 100,000 or more copies spread throughout the parasite genome (*Ramírez et al., 2015*). We reasoned that the spread of Cq values at low DNA input might reflect an unequal distribution of linked PCR target sequences in the aliquots and that fragmentation of the DNA could more evenly distribute the target sequences throughout the blood DNA sample, and thus increase the frequency of positive PCR reactions. Comparison of fragmented and non-fragmented parasite-spiked blood DNA samples confirmed this hypothesis, shifting the 100% positive PCR reaction one order of magnitude more sensitive (from $10^{-3}$ to $10^{-4}$ PE per reaction; *Figure 4A*). Fragmentation by either Covaris ultrasonication or a cuphorn attachment to a standard sonicator proved equally effective in increasing assay sensitivity (*Figure 4—figure supplement 1*).

To determine if DNA fragmentation also increased the sensitivity of detection of *T. cruzi* DNA in blood samples from infected hosts, we resampled 6 macaques, focusing on those that had been the most difficult to detect in the initial 1-year sampling study (*Figure 3*, *Figure 3—figure supplement 1*). Samples were collected and processed similarly as shown in *Figure 2A* except that 'C' samples were processed for DNA extraction and then fragmented and compared to the unfragmented samples A and B. As with the DNA-spiked samples, prior DNA fragmentation increased the sensitivity of PCR detection; fragmented samples from four of the six macaques had 80–100% positive reactions using only five replicates, indicating that a single PCR reaction could have detected most of these as *T. cruzi* positive while the unfragmented samples had frequencies of positive amplification from 0 to 40% (*Figure 4B*). Fragmentation was also beneficial in animals with higher parasite loads, providing 100% positive PCRs when fragmented compared to as low as 20% in the same non-fragmented samples (*Figure 4—figure supplement 2*).

## Replicate sampling, but not time between samples, facilitates detection of *T. cruzi* infection in hosts with the lowest parasite burden

Altogether, these results demonstrate that deep-sampling PCR can consistently detect *T. cruzi* DNA in the blood of *T. cruzi*-infected macaques well below the $10^{-3}$ level (equivalent to ~0.5 parasites/ml of blood) typically considered the limit of quantification in *T. cruzi* PCRs (*Schijman et al., 2011*) and that prior fragmentation can reduce by 10× the number of replicate reactions needed to detect a positive. However, animals with low and/or infrequent circulating parasite DNA still resist detection at a single bleed point (e.g. *Figure 4*; macaques P2 and P11); indeed, a number of macaques were negative by

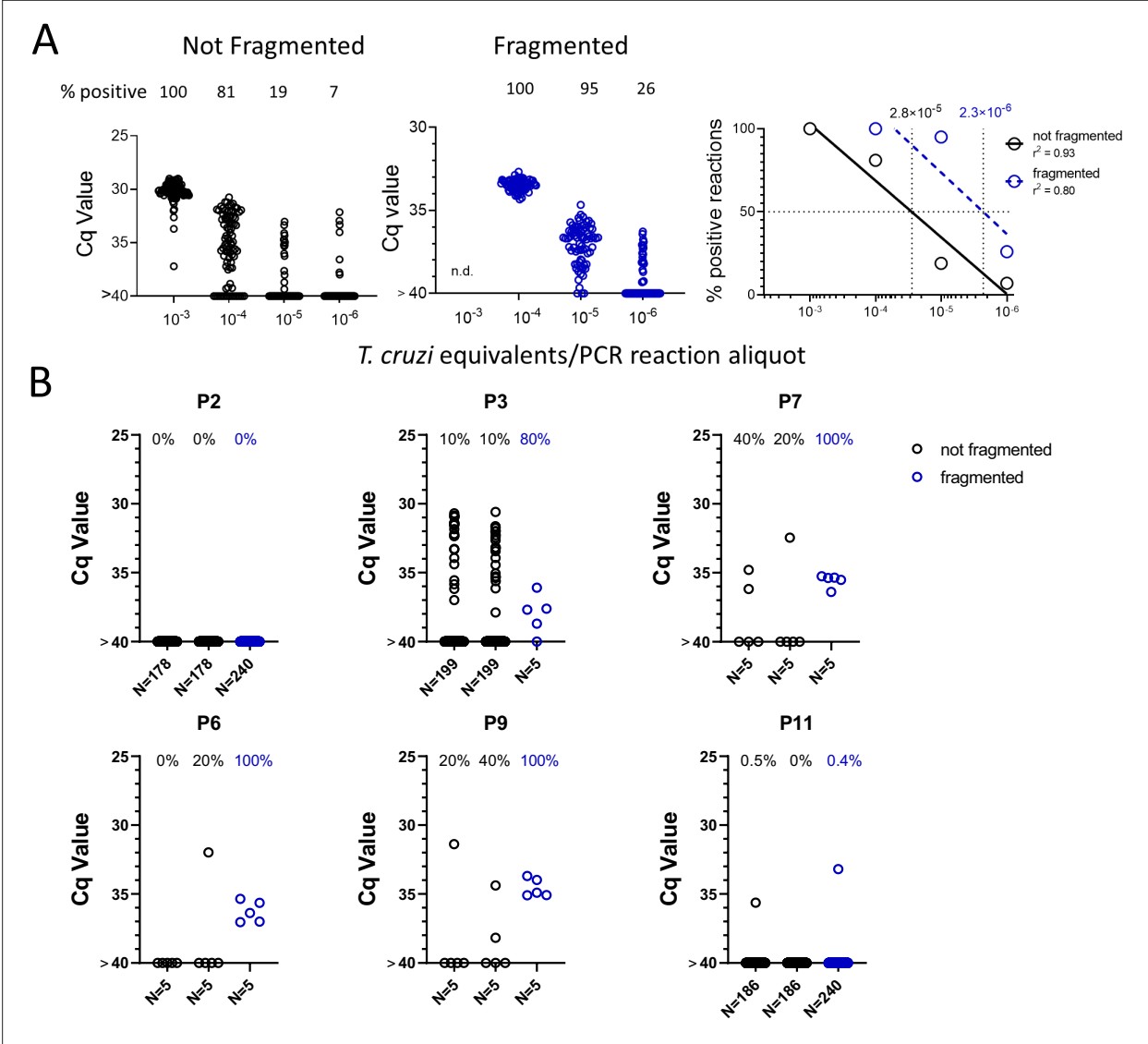

**Figure 4.** DNA fragmentation increases the sensitivity of PCR detection of *T. cruzi* DNA. (**A**) The quantity of parasite DNA required to yield 50% positive PCR reactions in replicate assays is reduced by >10-fold by cuphorn fragmentation of DNA from blood. A total of 72–96 replicate PCR reactions were conducted for each sample. (**B**) The increased sensitivity of consistent detection of *T. cruzi* DNA achieved by prior fragmentation is evident in samples from infected macaques. Blood samples were collected as described in *Figure 2A* for samples A and B but in this case, sample C was also used for DNA isolation and that DNA fragmented by cuphorn sonication. *N* = the number of replicate PCR.

The online version of this article includes the following source data and figure supplement(s) for figure 4:

**Source data 1.** Source data for *Figure 4A*.

**Source data 2.** Source data for *Figure 4B*.

**Figure supplement 1.** Fragmentation of DNA increases the frequency of positive replicate PCR reactions in a DNA sample.

**Figure supplement 1—source data 1.** Source data for *Figure 4—figure supplement 1C, D*.

**Figure supplement 1—source data 2.** Image files of original gels for *Figure 4—figure supplement 1*.

**Figure supplement 1—source data 3.** Marked image files of original gels for *Figure 4—figure supplement 1*.

**Figure supplement 2.** DNA fragmentation results in more consistent PCR amplification and detection, even in samples where *T. cruzi* DNA may be detectable using a single PCR assay.

**Figure supplement 2—source data 1.** Source data for *Figure 4—figure supplement 2*.

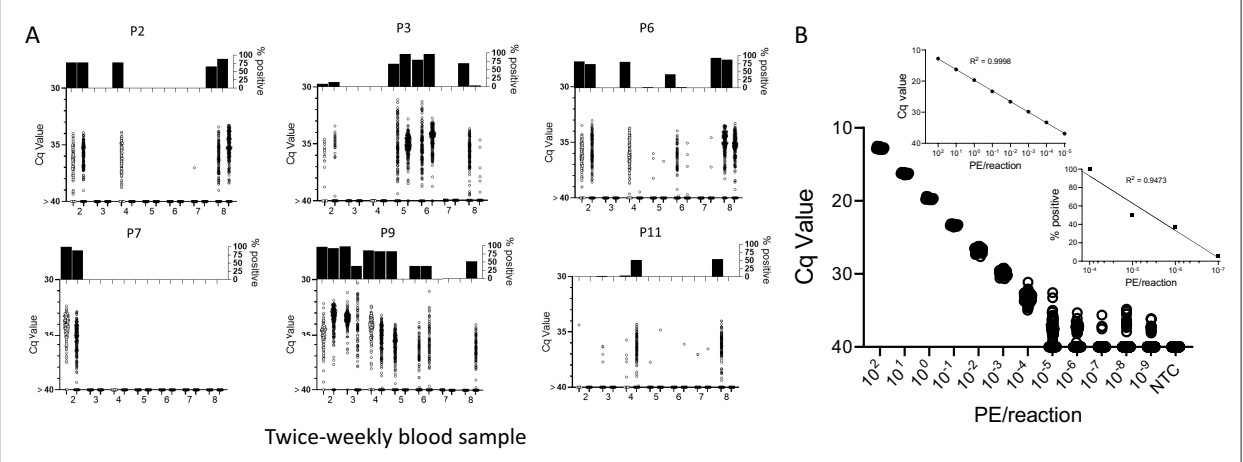

**Figure 5.** Repeat blood sample collection and deep-sampling PCR of fragmented DNA can quantify infection load over a minimum of eight orders of magnitude. (**A**) Frequent blood sampling demonstrates the sampling error involved in the detection of *T. cruzi* DNA in infected macaques with low parasite burden. Duplicate blood samples were collected twice weekly for 4 weeks from six macaques with the lowest overall PCR positive rate in the monthly sampling study (*Figure 3*, *Figure 3—figure supplement 1*, and *Supplementary file 1*). Bleed 1 was used in the experiments in *Figure 4*; the results of bleeds 2–8 are shown here. DNA was extracted from the duplicate samples at each bleed point and subjected to fragmentation before aliquots were used in 184 replicate PCR reactions/sample. The bottom of each subfigure shows the Cq value of each replicate reaction and the top plots the % positive reactions. (**B**) Replicate PCR analysis of fragmented macaque blood DNA spiked with known parasite equivalents (PE) of *T. cruzi* DNA. Insets show the linear relationship between Cq values and PE/reaction over the high range of PE and percent positive reactions and PE/reaction on the lower range of inputs. 10–388 replicate reactions were conducted for each dilution. NTC = no *T. cruzi* DNA.

The online version of this article includes the following source data for figure 5:

**Source data 1.** Source data for *Figure 5A*.

**Source data 2.** Source data for *Figure 5B*.

deep sampling PCR on duplicate blood samples at greater than half of the monthly sampling points. To determine if this variation in detection at different sampling times was random (due primarily to sampling error) or reflected true fluctuations in parasite load over time, the six macaques shown in *Figure 4* were sampled an additional seven times over 4 weeks, and replicate fragmented blood DNA samples submitted to deep-sampling PCR.

As shown in *Figure 5A*, blood sampling over this short time frame gave a similar irregular pattern of DNA detection as did the monthly samples, with some animals >80% positive for replicate PCRs at some points and 0% positive on multiple others (e.g. P2 and P7). It was also striking that in multiple animals, relatively high PCR positive rates were observed in one of the samples but not in the duplicate sample collected at the same time (indeed usually using the same needle stick; e.g. P2 4; P9 bleeds 5 and 8; and P11 bleed 8). These results emphasize the random sampling error involved in detecting *T. cruzi* DNA in hosts with very low parasite burden. However, collectively, the overall frequency of positive PCR reactions for these 6 macaques in year 2 (8, twice weekly collections; fragmented DNA) compared to the same animals in year 1 (11–12 monthly collections; non-fragmented DNA; Table S1) was >10-fold higher (2.2% vs. 27.5%).

To determine the full range of quantification of *T. cruzi* DNA in blood that is possible using both DNA fragmentation and deep sampling, we analyzed blood spiked with *T. cruzi* DNA over 12 orders of magnitude (*Figure 5B*). As noted above, DNA fragmentation alone extends the range over which a single PCR reaction is dependably positive to $10^{-4}$ PE/reaction (*Figure 4*). Below $10^{-4}$ PE/aliquot, both the average Cq value and the percent positive reactions allow quantification to $10^{-5}$ PE and the percent positive reactions remains linear to $10^{-7}$ PE/assay. Thus, the combination of fragmentation and deep-sampling extends the quantifiable range of detection of *T. cruzi* by four orders of magnitude below previous estimates, to ~2.4 × $10^{-5}$ parasites/ml of blood, or fractions of a parasite in blood DNA. This detection level is also consistent with the known frequency of the target satellite DNA repeat (approximately $10^5$/parasite) and suggests that the assay is approaching single copy DNA detection/reaction. Detection of *T. cruzi* is possible well below this limit of quantification but is

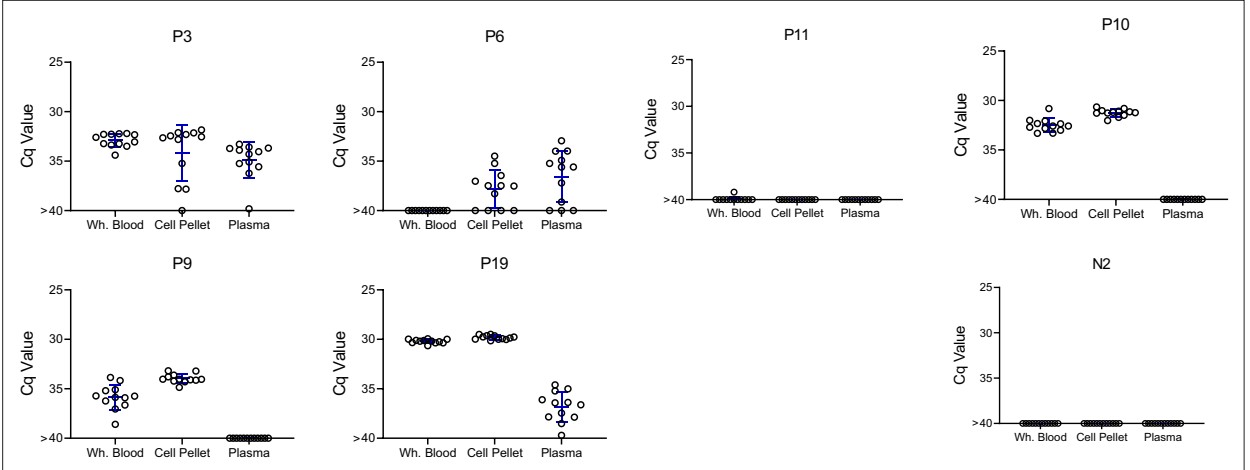

**Figure 6.** *T. cruzi* DNA can be detected in whole blood, the cell pellet of heparinized blood, or plasma, but is at the highest and most consistently detected in the blood cell pellet. Replicate blood samples taken at the same time were either processed as whole blood (sample 1) or separated into the cell pellet and plasma (sample 2) by centrifugation, for DNA purification and fragmentation. Twelve replicate aliquots of 125 ng of DNA each were amplified by PCR for each fraction. Mean and standard deviation are shown.

The online version of this article includes the following source data for figure 6:

**Source data 1.** Source data for *Figure 6*.

essentially random and is dependent on the number of blood samples and their volume collected, as well as the number of replicate PCR reactions conducted on each.

## The blood cell pellet routinely contains the highest amount of detectable parasite DNA relative to other blood fractions

Several recent studies have successfully used patient serum or plasma as a source for detection of *T. cruzi* DNA (*Kann et al., 2020*; *Kann et al., 2023*). To directly determine the optimal source of DNA for parasite detection, we compared DNA extracted from whole blood, the cell pellet from blood, or plasma. In all cases, 125 ng of the source DNA was used for each replicate PCR. Although all DNA sources provided positive replicates for one or more animals, both plasma DNA and whole blood DNA had all negative replicates in several cases when the replicates of DNA from the cell pellet were majority positive (*Figure 6*). Thus, plasma is a useful, but probably less dependable source for detection of *T. cruzi* DNA in the blood of chronically infected macaques.

## Deep-sampling PCR demonstrates that humans and dogs exhibit a similar range of *T. cruzi* burdens as macaques and detection of DNA in blood predicts DNA detection in tissues

The deep sampling approach optimized using samples from naturally infected macaques performs similarly with blood samples from humans and dogs (*Figure 7*) and both humans and dogs exhibited the same broad range of detectable parasite DNA in circulation across individuals as observed in the macaques. Although it is rare to be able to compare parasite detection in blood or plasma with that in tissues of the same animal, we had that opportunity in three *T. cruzi*-infected animals, a dog which succumbed to heartworm infection, and two macaques who were euthanized for health reasons unrelated to *T. cruzi* infection. As expected, animals with detectable blood parasite DNA also had PCR-amplifiable DNA in samples from multiple tissues (*Figure 7—figure supplement 1*).

Lastly, we used the deep sampling PCR approach to monitor outcomes in dogs under treatment using benznidazole in a high-dose/less frequent dosing protocol which has shown variable efficacy in multiple species (*Bustamante et al., 2023*; *Bustamante et al., 2020*). Treatment monitoring by deep-sampling PCR in three dogs with different treatment outcomes is shown in *Figure 8*. TFu1 was infected as a puppy and one of his littermates succumbed to *T. cruzi* infection before treatment could be initiated (*Lim et al., 2024*). Twice weekly dosing for nearly 12 months reduced, but did not totally clear parasites from blood. Increasing the dosing frequency to 3 times per week resulted

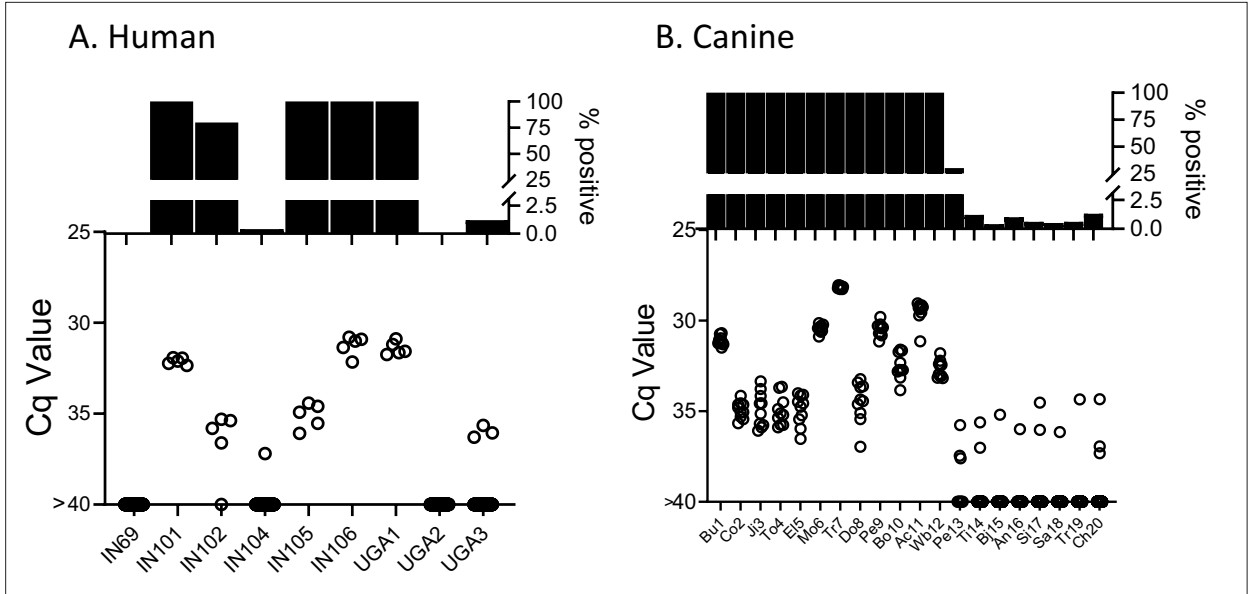

**Figure 7.** Deep sampling PCR of (**A**) fragmented DNA from whole blood from 9 not-treated, chronically infected humans and (**B**) fragmented blood cell pellet DNA from 20 seropositive dogs. An additional 10 seropositive dogs from this study group were negative for up to 384 replicate PCR reactions (not shown).

The online version of this article includes the following source data and figure supplement(s) for figure 7:

**Source data 1.** Source data for *Figure 7A*.

**Source data 2.** Source data for *Figure 7B*.

**Figure supplement 1.** Detection of *T. cruzi* infection by deep-sampling PCR of blood- or plasma-derived DNA is corroborated by the PCR detection of *T. cruzi* DNA in individual tissue samples from skeletal muscle, heart, or other organs including liver, spleen, and gut.

**Figure supplement 1—source data 1.** Source data for *Figure 7—figure supplement 1*.

in undetectable parasites. Twice weekly dosing in TDa2 also initially appeared to be successful but detection of *T. cruzi* DNA in samples collected at 10 and 28 weeks prompted a decision to change the dosing frequency to 3 times per week. TPe3 still had robust levels of *T. cruzi* DNA at 1 week post-treatment but was negative by deep-sampling PCR while on a twice weekly dosing regimen. Proof of cure in these dogs will require evidence of continued negative deep-sampling PCR after the end of treatment, but these results demonstrate that, as expected based upon many studies in humans, the response to BNZ treatment can be variable between individuals but that rigorous assessment of

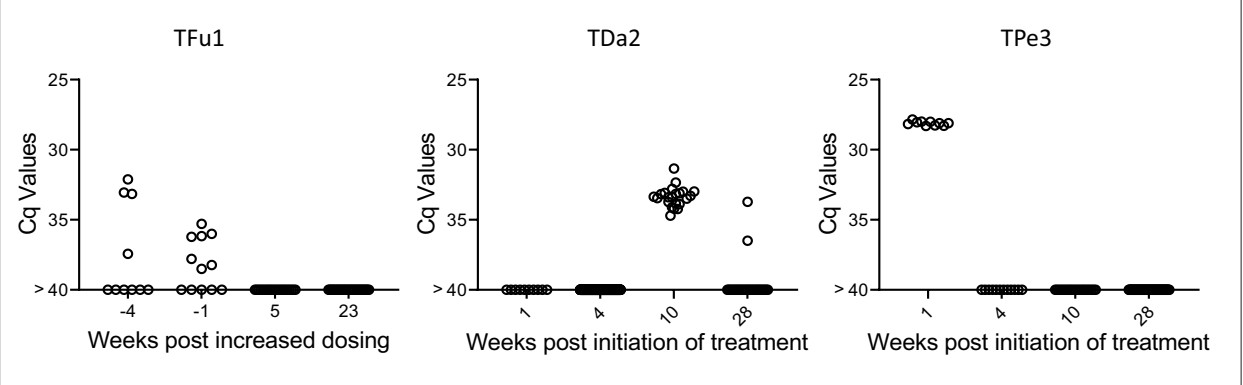

**Figure 8.** Monitoring by deep sampling PCR of *T. cruzi* DNA from whole blood in dogs during treatment with benznidazole. Dogs were treated with 18–20 mg/kg benznidazole twice (initially) or thrice (if remaining PCR positive) per week.

The online version of this article includes the following source data for figure 8:

**Source data 1.** Source data for *Figure 8*.

treatment effectiveness during the treatment course allows adjustments to dosing that may provide higher overall treatment success.

## Discussion

Without question, the inability to detect and quantify *T. cruzi* in blood or other samples from individuals with suspected infection, is the key impediment to identifying candidates for treatment, determining if the treatment has been effective in resolving the infection, and validating potential new treatments. Additionally, without the ability to routinely quantify parasite load in humans and other naturally infected hosts, there are multiple, sometimes contentious issues in Chagas disease that will remain unresolved, among these, the relative role of parasite and host genetics in establishing the parasite burden in this persistent infection, the stability of parasite load over time, and particularly important, the relationship between parasite burden and the presence and severity of clinical disease.

Quantifying *T. cruzi* in blood and other tissues in hosts after the early peak of infection is challenging because the immune response to *T. cruzi* infection is normally extremely effective in tightly limiting parasite numbers within a few months of the initial infection. The bulk of *T. cruzi* in vertebrate hosts at any point in time are intracellular, mostly in various muscle types throughout the body. However, to maintain and to transmit the infection, *T. cruzi* must also exit host cells (following a 4- to 5-day period of parasite multiplication) and infect new host cells or on occasion, be ingested by blood-feeding triatomines. In addition to intact parasites, the blood is likely to contain DNA released by parasites that are killed by immune effector mechanisms and thus may be available for detection, either free in plasma or bound to blood cells (*Lam et al., 2021*; *Thompson et al., 2024*). In short, even in hosts with very low-level/well-controlled infections, the potential always exists for *T. cruzi* DNA to be present in the blood, albeit often at extremely low amounts. And when present in such low quantities, the chances of detection, even with highly optimized PCR protocols, is random and highly prone to sampling error. This is not a situation unique to *T. cruzi* infection. For example, detection of SARS-CoV-2 (*Waked et al., 2020*) and HIV (*Rutsaert et al., 2018*) using PCR is also prone to failure when employing single amplification reactions. The ability of replicate PCR reactions to overcome this limitation has been discussed in depth in PCR optimization reviews (*Taylor et al., 2019*) and one study (*Stowers et al., 2010*) reported that by averaging hundreds of replicate reactions one can vastly extend the concentration range over which PCR can provide reliable detection of DNA that is in low abundance. However, the use of hundreds of replicate PCR reactions as conducted here has been rarely, if ever, used to routinely validate the presence of any infectious agent in blood.

It is well documented that *T. cruzi*-positive hosts give variable results in PCR tests, and this variability increases with decreasing parasites or parasite DNA in a sample. This is primarily due to subsampling error (*Taylor et al., 2019*) and this error functions at two levels when screening blood for the presence of pathogen DNA. First, a blood sample is a subsample of the entire blood volume of a potential host; e.g. a 10-ml blood sample is ~0.2% of the total volume of blood in an average adult human. The existence of this subsampling error in *T. cruzi* infection is well illustrated by the fact that two replicate blood samples from macaques collected using the same needle stick can yield a substantial number of positive PCR reactions in one sample, and zero positive reactions in the other (*Figure 5*). Simply collecting samples at different times, whether a few days (*Figure 5*) or a few months (*Figure 3*) apart, does not necessarily solve this problem. When parasite numbers are very low, the chance that a small subsample of the total blood will contain parasite DNA is random irrespective of when it is collected. The only solution is to obtain more or larger blood samples so as to increase the chances of collecting a fragment of the parasite DNA that is in the circulation. This same subsampling error acting at the level of whole parasites was evident in the xenodiagnosis studies of Cerisola, wherein it was rare to obtain parasite-positive bugs when they fed on certain subjects who apparently had very low levels of parasites circulating in the blood (*Cerisola, 1977*).

A second opportunity for subsampling error under low target conditions occurs when an aliquot of the total blood DNA obtained in a blood draw is used for PCR amplification. In the case of the ~5-ml macaque blood samples used in the bulk of the assays in this study, from 1/100th to 1/250th of the total recovered DNA was used for each replicate PCR reaction. In a number of animals, the entire blood DNA sample was exhausted doing replication PCR reactions without a positive reaction while in most animals, some DNA remained after nearly 200, sometimes all negative, amplification reactions/sample. We show that fragmentation of DNA significantly decreases the subsampling error

at this level, presumably by breaking the DNA containing one or more target satellite DNA regions into smaller fragments that are then more widely dispersed in the total blood DNA. DNA fragmentation led to an increased frequency of positive replicate PCR reactions and generally more consistent Cq values between these aliquots. Fragmentation increased amplification consistency and the sensitivity of the PCR protocol by one order of magnitude and should be incorporated as a standard step for amplification of satellite DNA target in *T. cruzi* even when deep-sampling is not used. However ultimately, the limiting factor in the consistent detection of *T. cruzi* in blood is the total amount of blood collected and processed for fragmentation and amplification. If there is no parasite DNA in a particular sample, no amount of fragmentation or number of amplification reactions will detect it. Also, in addition to sampling errors, the inefficiency of PCR to amplify very low abundance targets, particularly among a complex DNA mixture as exists in blood (termed the Monte Carlo effect; *Karrer et al., 1995*; *Bustin and Nolan, 2004*) may also contribute to the failure to detect target *T. cruzi* DNA in some samples despite its presence.

In addition to providing much greater sensitivity for detecting active infection relative to conventional single or low replicate (2–3/sample) PCR analysis, conducting 100s of replicate PCR reactions on fragmented blood DNA also provides information on the relative abundance of parasites in the blood of infected hosts. Most implementations of PCR for *T. cruzi* have a limit of consistent detection of ~$10^{-3}$ PE/assay, which equates to ~0.5 parasites/ml of blood in our assays. Combining sample DNA fragmentation and deep-sampling extends the range of quantification to at least $10^{-6}$ PE per aliquot, or ~0.00025 parasites/ml (1 parasite per 4 l) of blood.

This considerably increased sensitivity of infection detection and relative quantitation made possible through the use of deep sampling PCR reveals the previously undocumented range of parasite burden between individuals with chronic *T. cruzi* infections. Although best demonstrated in the frequently sampled macaques in this work, both humans and dogs also exhibit a range of parasite burdens exceeding five orders of magnitude when compared to a standard curve (e.g. from Cq values of ~25 in the highest macaque and dog fragmented samples to <1% frequency of positive PCR reactions in some members of all three species). The similarity in the ranges of parasite load in the three species examined in this study again reinforces the similarities between *T. cruzi* infection in these species and the appropriateness of dogs and NHPs as models of the human infection (*Tarleton et al., 2024*).

In addition to the sampling error discussed above, the variation in the number of satellite DNA repeats per genome among *T. cruzi* isolates (*Schijman et al., 2011*) makes absolute quantitation of parasite load virtually impossible. However, such quantitation is not necessary in order to conclude that individuals with long-established *T. cruzi* infections control these infections to vastly different degrees. This ability to more precisely quantify parasite load should now allow investigations into the potential immune mechanisms that might be responsible for this variable control and assessment of the potential correlation between relatively stable parasite load and the chances of having or developing clinical disease – which is detected in <50% of long-term infected subjects.

This study also confirms across a broader range of individuals, the relative stability of parasite load over time, an expected phenomenon that was also apparent in the xenodiagnosis studies by *Cerisola, 1977*. A surprising finding was that a subset of macaques infected for 2–3 years and presenting a relatively higher parasite burden exhibited a significant decline in parasite load over the 1-year survey period. This result suggests that immune control mechanisms not only continue to confine parasite load but can in some cases also drive that load lower over time during the chronic phase of infection. However, this is not the pattern in all individuals, as some macaques with equally short-term infections have already restricted parasite numbers to nearly undetectable levels while some with longer-term infections maintain relatively higher parasite burden. It is also noteworthy that parasite control can be quickly lost when the health status (and presumably immune status) changes, as in the case of macaque P4, similar to what is observed in humans with suppressed immunity (*Lattes and Lasala, 2014*). These findings emphasize that even subjects controlling *T. cruzi* burden to very low levels should be considered for anti-parasite treatment as the immune control of the infection can be quickly lost.

The original goal of this study was to develop a 'test of cure' for use in the evaluation of clinical trials of anti-*T. cruzi* compounds. Previous candidate compounds used in clinical trials have failed to provide sterile cure and these failures were relatively easy to detect without a highly sensitive PCR protocol as described here (*Tarleton, 2023a*; *Tarleton, 2023b*). With the progression of highly promising new

compounds, including one that provided 100% sterile cure in NHPs with naturally acquired *T. cruzi* infection (*Padilla et al., 2022*), an assay that detects success rather than only failure is needed. The combination of DNA fragmentation and deep-sampling of replicate blood samples should fulfill that need. Operationally, this would not require any special equipment but does necessitate extreme care in sample preparation and assay execution. Also, conducting hundreds of PCR reactions on the DNA from 10 or more high volume (e.g. 10 ml or greater) blood samples from each subject will be expensive, time-consuming and labor-intensive. It is also worthy of note that in cases of treatment failure, the apparent abundance of *T. cruzi* DNA in the blood often return to pre-treatment levels (*Parrado et al., 2019*), so selection of subjects with more readily detectable pre-treatment parasite loads for such trials would make it relatively easy to detect treatment failures and provide greater confidence that those that remain negative following deep-sampling are indeed cured.

There may also be opportunities for additional improvements in PCR-based detection of *T. cruzi*. One of the limitations of this study is that we have so far only examined samples from a small number of human subjects. However, those samples fall into the same pattern of wide-ranging parasite loads as the macaque and dog samples so there is little reason to think that more extensive testing of human samples will reveal any surprises. Additional targets for PCR amplification have been identified (*Kann et al., 2020*; *Kann et al., 2023*) and could be multiplexed in a single reaction to perhaps achieve greater sensitivity. We also may not have exhausted the limits of DNA fragmentation for dispersing target DNA in a blood sample in order to further reduce subsampling errors or explored the tolerance for increased loading of fragmented DNA for each PCR reaction. Improvements in any of these areas could reduce the number of replicate PCR reactions without compromising sensitivity. Finally, new technologies such as UltraPCR, which allows for higher DNA loading, extensive multiplexing, and the generation of >30 million individual PCR reaction per tube (*Lai et al., 2023*; *Shum et al., 2022*) could greatly reduce the number of replicate PCR assays that are needed for high sensitivity detection. Such developments could make the PCR-based detection of all *T. cruzi*-infected subjects possible, if not routine.

## Materials and methods
### Macaques
All NHP utilized for these studies were part of the approximately 1000-animal, Rhesus Macaque (*Macaca mulatta*) Breeding and Research Resource housed at the AAALAC accredited, Michale E. Keeling Center for Comparative Medicine and Research (KCCMR) of The University of Texas MD Anderson Cancer Center in Bastrop. TX. This is a closed colony, which is specific pathogen free for Macacine herpesvirus-1 (Herpes B), Simian retroviruses (SRV-1, SRV-2, SIV, and STLV-1), and *Mycobacterium tuberculosis* complex. Study animals that were seropositive for *T. cruzi* had acquired the infection naturally through exposure to the insect vector of the parasite while in their indoor–outdoor housing facilities. The NHP experiments were performed at the KCCMR and all protocols were approved by the MD Anderson Cancer Center's IACUC (ACUF# 00002241-RN00 and 00000451-RN03), and followed the NIH standards established by the Guide for the Care and Use of Laboratory Animals (*National Research Council, 2011*).

A total of 26 rhesus macaques that had been confirmed to be serologically positive for *T. cruzi* infection (a subset of which had been PCR positive in previous screenings; *Hodo et al., 2018*; *Supplementary file 1*) were utilized in these studies. Five of these macaques were previously treated and cured of *T. cruzi* infection in 2018 and consistently PCR negative since (*Padilla et al., 2022*). These cured macaques and two seronegative macaques served as control, uninfected animals. Under light injectable anesthesia, three ~5-ml peripheral blood samples (total ~15 ml) were collected from each animal and shipped overnight on ice packs for each sampling point. Except as indicated, whole blood was used for DNA extraction.

### Canines
Dogs used in these studies came from a network of kennels in central and south Texas with a history of triatomine vector occurrence and canine Chagas disease as previously described (*Bustamante et al., 2022*). At these large kennels, dogs are primarily bred and trained to aid hunting parties and the predominant breeds include Belgian Malinois, Brittany spaniels, cocker spaniels, English pointers,

German shorthaired pointers, Kelpies, Labrador retrievers, and hound dogs. Dogs >2 months of age, including males and females were sampled. Approximately 3 ml of blood was collected via jugular venipuncture into heparinized tubes which were centrifuged at 2000 × *g* for 15 min and the cell pellet and plasma separated before overnight shipment on ice. *T. cruzi* seropositive dogs were identified by *T. cruzi* multiplex serology (*Bustamante et al., 2022*) before blood cell pellet DNA testing by PCR. Some dogs previously confirmed as infected were treated using a twice-weekly high-dosing protocol as previously described (*Bustamante et al., 2023*).

Informed consent was obtained from dog owners prior to their participation, and this study was approved by the Texas A&M University Institutional Committee on Animal Use and Care and the Clinical Research Review Committee (IACUC 2018-0460 CA and IACUC 2022-0001 CA).

### Human

Nine subjects with positive serological findings for *T. cruzi* infection (i.e., positive in ≥2 of the 3 tests performed, indirect immunofluorescence assay, hemagglutination, and enzyme-linked immunosorbent assay; as previously described; *Castro Eiro et al., 2021*) were enrolled at the Hospital Interzonal General de Agudos Eva Perón, in Buenos Aires, Argentina. All participants had no signs of cardiac disease as revealed by electrocardiography and echocardiography testing. The protocol was approved by the Institutional Review Board (IRB) of Hospital Interzonal General de Agudos Eva Perón (Memorandum 19/19) of the Province of Buenos Aires, Argentina. Signed informed consent for sample use and publication of results was obtained from all individuals included in the study. Ten milliliters of blood were drawn from seropositive subjects by venipuncture into heparinized tubes (Vacutainer; BD Biosciences) and centrifuged at 1000 × *g* for 15 min. The plasma was collected in a separate tube and the blood cell pellet and plasma were frozen at −20°C and the de-identified samples shipped to the University of Georgia on dry ice.

### Blood DNA extraction and qPCR

DNA was extracted from all blood samples or fractions thereof (whole blood, pellet, plasma layer) using the Omega E.Z.N.A Blood DNA MAXI kit. Following the manufacturers protocol for 'up to 10 ml whole blood', approximately 2–5 ml of blood or blood fraction was lysed and processed bringing the initial volume of each sample up to 10 ml with the addition of PBS (Gibco 10010023) and eluting with 500 µl elution buffer. Samples that were frozen upon receipt (human samples) were thawed on ice before starting protocol. DNA samples were quantified with the nanodrop 2000 system (Themo Scientific) before diluting to 25 ng/µl in water (Invitrogen AM9937).

The qPCR assay used to detect *T. cruzi* DNA in blood in this study is essentially as previously described (*Padilla et al., 2022*) and used extensively in many labs, including those involved in clinical trial monitoring (*Sulleiro et al., 2019*). In brief, each reaction includes 125 ng of genomic DNA, 1 pg of internal amplification control (IAC) fragment, 0.75 µM of each *T. cruzi* satellite DNA- and IAC-specific primers, 0.5 µM each of *T. cruzi* satellite DNA and IAC probes and 10 µl of Bio-Rad iTaq Universal Probes Supermix and water to make a 20-µl final volume reaction. IAC template, primer, and probes are omitted from deep-sampling assays following the first 5–10 replicates. We used 384-well hard-shell plates (Bio-Rad # HSP3805) and Microseal 'B' Adhesive Seals (Bio-Rad #MSB1001) compatible with qPCR assays and the CFX Opus 384 real time PCR detection system under the following cycling conditions: (1) initial denaturation, 95°C, 3 min; (2) denaturation, 95°C, 15 s; (3) annealing, 58°C, 1 min; (4) 50× cycles. A standard sample with a known concentration of *T. cruzi* DNA ($5.6 \times 10^{-3}$ PE) is included in each plate for reference. Analysis of the data was done using CFX Maestro software version 2.3 (Bio-Rad). For samples with no amplified product (no Ct value can be calculated) the Ct value is set to '40' for graphing purposes.

A step-by-step protocol is available at https://www.protocols.io/, DOI: https://dx.doi.org/10.17504/protocols.io.ewov1dm87vr2/v1.

### Preventing contamination

To prevent contamination when preparing high numbers of qPCR reactions, a number of safeguards were employed. The setup for all reactions was performed in a space separate from our main research laboratory and in a class II A2 laminar flow hood otherwise not used for work with *T. cruzi* and using supplies and instruments that were specific for this lab space. All components are sprayed with 70%

ethanol and DNA decontamination spray (LookOUT, Sigma) before being brought into the hood and all hood surfaces are sprayed with ethanol and decontaminant at the end of each work session, followed by UV exposure. PCR assay components were aliquoted from large batches into individual tubes sufficient for each 384-well plate. The centrifuge used for DNA extractions was bleached and decontaminated before processing the final elution for each DNA batch. DNA samples are diluted into 96-well plates and then loaded 96 wells at a time into the 384-well plate using the mini 96 pipette (Integra Mini 96). Despite these procedures, one set of DNA isolations from *T. cruzi*-negative controls showed evidence of cross-contamination when they were all processed at the same time as a set of DNA standards containing high quantities of *T. cruzi* DNA. These 'February' samples for seven animals were excluded from further analysis. Otherwise, samples from animals previously cured of *T. cruzi* infection by treatment with AN15368 (*Padilla et al., 2022*) processed at the same time as other 'infected' samples showed no evidence of cross-contamination.

## DNA fragmentation

Purified DNA from blood was fragmented by either in a Covaris E220 Focused-ultrasonicator (which allowed precise selection of target DNA fragment sizes of 1000, 500, or 300 bases), or using the 3 inch cuphorn attachment to a Branson SFX250 Sonifier cooled by a Bio-Rad 1000 mini chiller circulator. For cuphorn sonication, DNA samples (typically 500 µl in volume and 15–150 µg total DNA) in 1.5 ml Eppendorf snap cap tubes were wrapped securely in parafilm and kept on ice before fragmentation. Sonifier settings were set to continuous mode with a time interval of 50 s and 60% amplitude for five cycles.

## PCR standards

Samples containing known amounts of parasite DNA were generated using 5 ml whole blood with $10^7$ *T. cruzi* epimastigotes of the Brazil strain added. The blood was then treated the same as experimental blood with DNA extracted as described with the Omega MAXI kit. The DNA was diluted to 25 ng/µl and serially diluted 10-fold with 25 ng/µl naive blood DNA.

## Hemoculture

Peripheral blood (~5 ml) was aliquoted into five replicate T25 flasks for incubation at 26°C in supplemented liver digest neutralized tryptose medium as described previously (*Padilla et al., 2021*). The presence of *T. cruzi* parasites was assessed every week for up to 3 months under an inverted microscope.

## Tissue PCR

Single tissue samples (8 mm biopsies) or pooled samples from five sites totaling ~500 µl per pool were obtained from frozen necropsied tissues, processed and subjected to PCR amplification of *T. cruzi* DNA as previously described (*Padilla et al., 2022*).

## Acknowledgements

Special thanks to Stephanie Collins, MS, DVM for field collection of dog samples and Ty McAdams and Luke Segura from KCCMR for assistance with coordination of NHP sampling. We thank the staff and patients of the Hospital Interzonal General de Agudos Eva Perón, Buenos Aires, Argentina who provided blood samples.

## Additional information

### Funding

| Funder | Grant reference number | Author |
| --- | --- | --- |
| National Institutes of Health | R03 AI166504 | Rick L Tarleton |

| Funder | Grant reference number | Author |
|---|---|---|
| National Institutes of Health | 1UL1TR003163 | Sarah Hamer<br>Ashley B Saunders |

The funders had no role in study design, data collection, and interpretation, or the decision to submit the work for publication.

## Author contributions

Brooke E White, Investigation, Writing – original draft, Writing – review and editing; Carolyn L Hodo, Resources, Investigation; Sarah Hamer, Ashley B Saunders, Susana A Laucella, Resources; Daniel B Hall, Formal analysis; Rick L Tarleton, Conceptualization, Supervision, Funding acquisition, Visualization, Methodology, Writing – original draft, Project administration, Writing – review and editing

## Author ORCIDs

Rick L Tarleton  https://orcid.org/0000-0002-9589-5243

## Ethics

Nine subjects with positive serological findings for T. cruzi infection were enrolled at the Hospital Interzonal General de Agudos Eva Perón, in Buenos Aires, Argentina. The protocol was approved by the Institutional Review Board (IRB) of Hospital Interzonal General de Agudos Eva Perón (Memorandum 19/19) of the Province of Buenos Aires, Argentina. Signed informed consent for sample use and publication of results was obtained from all individuals included in the study. Ten milliliters of blood were drawn from seropositive subjects by venipuncture into heparinized tubes (Vacutainer; BD Biosciences) and centrifuged at 1000 × g for 15 min. The plasma was collected in a separate tube and the blood cell pellet and plasma were frozen at −20°C and the de-identified samples shipped to the University of Georgia on dry ice.

The NHP experiments were performed at the AAALAC accredited, Michale E. Keeling Center for Comparative Medicine and Research (KCCMR) of The University of Texas MD Anderson Cancer Center and all protocols were approved by the MD Anderson Cancer Center's IACUC (ACUF# 00002241-RN00 and 00000451-RN03), and followed the NIH standards established by the Guide for the Care and Use of Laboratory Animals. Informed consent was obtained from dog owners prior to their participation, and this study was approved by the Texas A&M University Institutional Committee on Animal Use and Care and the Clinical Research Review Committee (IACUC 2018-0460 CA and IACUC 2022-0001 CA).

Reviewer #1 (Public review): https://doi.org/10.7554/eLife.104547.2.sa1
Reviewer #2 (Public review): https://doi.org/10.7554/eLife.104547.2.sa2
Author response https://doi.org/10.7554/eLife.104547.2.sa3

# Additional files

## Supplementary files

Supplementary file 1. Historical and year one PCR and hemoculture summary for macaques. N1 and N2 are seronegative controls and macaques T1–T5 were previously infected but cured of *T. cruzi* infection using benzoxaborole AN15368 (*Padilla et al., 2022*).

MDAR checklist

## Data availability

All data generated or analyzed during this study are included in the primary or supplemental data files provided in the manuscript. Tabulated raw data are provided in source data files.

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
