## [Editor Report · eLife Assessment]

This study presents an **important** methodological advance to improve the sensitivity of PCR for detecting Trypanosoma cruzi in blood, combining DNA fragmentation, deep sampling, and blood cell pellet analysis. The findings offer **solid** evidence of enhanced detection sensitivity and shed light on parasite load dynamics during chronic infection in mammalian reservoirs. The evidence is sound for macaques and the method shows promise in expanding detection limits, but there is some variability in the limits of detection and small sample size of human samples. This work will be of interest to parasitologists, epidemiologists, and clinicians using molecular diagnostics to monitor responses to etiological treatments for Chagas disease.

---

## [Referee Report · Reviewer #1 (Public review)]

This study presents a refined approach to enhance the sensitivity of PCR for detecting Trypanosoma cruzi in blood by employing DNA fragmentation and deep sampling, involving multiple replicate PCR reactions. Combined with serial blood sampling, these methods enabled consistent detection of the parasite in infected humans, non-human primates, and dogs, including hosts with very low parasitemia levels.

Inspired by earlier methods that cleaved kinetoplast DNA (kDNA) to improve target distribution, this study targets nuclear satellite DNA repeats, which are tandemly arranged in T. cruzi chromosomes. By fragmenting DNA prior to PCR, the authors reduced subsampling errors, breaking large fragments into smaller, evenly distributed units. This improved the frequency of positive reactions and reduced variability among replicate Cq values.

Using contrived blood samples, the study demonstrated that this approach significantly enhances PCR positivity. Moreover, the findings suggest that cell pellets from blood yield higher concentrations of parasite DNA compared to whole blood, prompting a reevaluation of current diagnostic practices, which predominantly use whole blood lysates.

The study also highlights the importance of deep sampling. Serial testing across multiple blood samples mitigated the variability in parasitemia, addressing challenges first noted in early xenodiagnosis studies (Cerisola et al., 1977).

The proposed DNA extraction and amplification procedures effectively captured parasitemia dynamics, achieving detection sensitivities with quantification limits as low as ~0.00025 parasite equivalents/mL, approaching the detection of a single target copy per reaction.

This work underscores the utility of deep-sampling PCR in monitoring parasitemia dynamics and guiding treatment strategies, especially in chronic infections. It also stresses the importance of treating individuals with low parasitic loads, as immune control may change over time.

Strengths:

The strategies used for increasing PCR sensitivity offer the potential for enhancing treatment monitoring and understanding the dynamics of parasite-host interactions in chronic Chagas disease.

Weaknesses:

While the study offers valuable insights for research in T.cruzi infection dynamics and monitoring of trypanocidal drugs efficacy, its broader adoption depends on the development of cost-effective and scalable alternatives to labor-intensive techniques such as sonication, currently required for DNA fragmentation. Additionally, the reliance on blood cell pellets and the DNA fragmentation protocol introduces extra processing steps, which may not be feasible for many clinical laboratories, particularly in resource-limited endemic areas that require simpler and more streamlined procedures.

---

## [Referee Report · Reviewer #2 (Public review)]

Summary:

This study introduces a valuable methodological innovation for detecting Trypanosoma cruzi, the causative agent of Chagas disease, using "deep-sampling PCR" which combines DNA fragmentation with multiple qPCR replications (>300 in some cases) on each sample. The authors aim to overcome the limitations of current qPCR methods by increasing the sensitivity of detection, which is fundamental for evaluating treatment responses in chronic Chagas disease patients. The work also evaluates the approach in multiple host species (macaques, humans, and dogs), at different times and across different sample types, including whole blood, blood cell pellets, plasma, and tissues.

Strengths:

The primary strength of this study lies in its methodological novelty, particularly the combination of multiple parallel PCR reactions and DNA fragmentation to enhance sensitivity. It is a sort of brute-force method for detecting the parasite. This approach promises the detection of parasitic DNA at levels significantly lower than those achievable with standard qPCR methods. Additionally, the authors demonstrate the utility of this method in tracking parasitemia dynamics and post-treatment responses in macaques and dogs, providing valuable insights for both research and clinical applications.

Weaknesses:

(1) Methodological Concerns on detection and quantification limits

Some methodological inconsistencies and limitations were observed that merit consideration. In Figure 1, there is a clear lack of consistency with theoretical expectations and with the trends observed in Figure 4A. Based on approximate calculations, having 10^-7 parasite equivalents with 100,000 target copies per parasite implies an average of 0.01 target copies per reaction. This would suggest an amplification rate of approximately 1 in 100 reactions, yet the observed 30% amplification appears disproportionately high. In addition, Figure 4A (not fragmented) shows lower values of positivity than Figure 1 for 10^-5 and 10^-6 dilutions showing this inconsistency among experiments. Some possible explanations could account for this inconsistency: (1) an inaccurate quantification of the starting number of parasites used for serial dilutions, or (2) random contamination not detected by negative controls, potentially due to a low number of template molecules.

Similarly, Figure 5B presents another inconsistency in theoretical expectations for amplification. The authors report detecting amplification in reactions containing 10^-9 parasites after DNA fragmentation. Based on the figure, at least 3 positives (as I can see because raw data is not available) out of 388 PCRs are observed at this dilution. Assuming 100,000 copies of satellite DNA per parasite, the probability of a single copy being present in a 10^-9 dilution is approximately 1/10,000. If we assume this as the probability of amplification of a PCR (an approximation), by using a simple binomial calculation, the probability of at least 3 positive reactions out of 388 is approximately 9.39 x 10^-6 (in ideal conditions, likely lower in real-world scenarios). This translates to a probability of about 1 in 100,000 to observe such frequency of positives, which is highly improbable and suggests either inaccuracies in the initial parasite quantification or issues with contamination. In addition, at 10^-6 PE/reactions (the proposed limit of quantification) it is observed that 40% of repetitions are amplified. The number of repetitions is not specified but probably more than 50 according to the graph. Such dilution implies 0.1 targets per reaction (assuming 100.000 copies divided by 10^6), which means a total of 5 target molecules to distribute among the reactions (0.1 targets multiplied by 50 reactions). It seems highly improbable that 40% of the reactions (20/50) would amplify under the described conditions. Even considering 200.000 target copies per parasite implies 0.2 targets per reaction and an average of 10 molecules to distribute among 50 reactions. The approximate probability of the observation of at least 20/50 positives can be calculated by determining the probability of a reaction to receive targets by assuming a random distribution of the targets among the tubes, p = 1 - (1 - 1/50)^10, and then by using a binomial distribution to determine the probability that at least 20 reactions receive at least one target copy. The probability of at least 20/50 positive reactions in a dilution of 10^-6 parasites (200.000 target copies per parasite) is 0.00028. Consequently, the observed result is highly unlikely.

1. Lack of details on contamination detection

Additionally, the manuscript does not provide enough details on how cross-contamination was detected or managed. It is unclear how the negative controls (NTCs) and no-template controls were distributed across plates, in terms of both quantity and placement. This omission is critical, as the low detection thresholds targeted in this study increase the risk of false positives by contamination. To ensure reliability and reproducibility, future uses of the technique would benefit from more standardized and clearly documented protocols for control placement and handling.

1. Unclear relevance for treatment monitoring in Humans

In Figure 7A, the results suggest that the deep-sampling PCR method does not provide a clearly significant improvement over conventional qPCR in humans. Of the 9 samples tested, 6 (56%) were consistently amplified in all or nearly all reactions, indicating these samples could also be reliably detected with standard PCR protocols. Two additional samples were detected only with the deep-sampling approach, increasing sensitivity to 78%; however, these detections might be attributable to random chance given the limited sample size. While the authors acknowledge the small sample size in the discussion, they do not address the fact that a similar increase in sensitivity was reported in citation 5, where only 3 samples were tested with 3 replicates each. This raises an important question: how many PCR reactions are needed in human samples to reach a plateau in detection rates? This issue should be further discussed to contextualize the results and their implications.

Despite these limitations, this work represents a promising step forward in the development of highly sensitive diagnostic tools for T. cruzi. It offers a novel foundation for advancing the detection and monitoring of parasitemia, which could significantly benefit Chagas disease research community and clinicians focused on neglected tropical diseases. While addressing the methodological inconsistencies and improving robustness will be critical, this study provides valuable insights and data that could lead to future innovations in parasitological research and diagnostics.

---

## [Author Response]

**Reviewer #1 (Public review):**
[…] Strengths:The strategies used for increasing PCR sensitivity offer the potential for enhancing treatment monitoring and understanding the dynamics of parasite-host interactions in chronic Chagas disease.Weaknesses:While the study offers valuable insights for research in T.cruzi infection dynamics and monitoring of trypanocidal drugs efficacy, its broader adoption depends on the development of cost-effective and scalable alternatives to labor-intensive techniques such as sonication, currently required for DNA fragmentation. Additionally, the reliance on blood cell pellets and the DNA fragmentation protocol introduces extra processing steps, which may not be feasible for many clinical laboratories, particularly in resource-limited endemic areas that require simpler and more streamlined procedures.

We agree that this methodology is likely to be used primarily as a research tool and for selective use in the field (e.g. drug trials) and unlikely to be standard in many clinical labs, irrespective of resources. We note the protocol does not require cell pellets (although that fraction provides the highest sensitivity) and that the fragmentation step is not at all labor-intensive. But to achieve consistent detection across the range of parasite burden known to occur in chronic *T. cruzi* infection, appropriately processed DNA from higher volumes of blood than are now routinely used for detection of *T. cruzi*, will be required.

**Reviewer #2 (Public review):**
[…] Strengths:The primary strength of this study lies in its methodological novelty, particularly the combination of multiple parallel PCR reactions and DNA fragmentation to enhance sensitivity. It is a sort of brute-force method for detecting the parasite. This approach promises the detection of parasitic DNA at levels significantly lower than those achievable with standard qPCR methods. Additionally, the authors demonstrate the utility of this method in tracking parasitemia dynamics and post-treatment responses in macaques and dogs, providing valuable insights for both research and clinical applications.Weaknesses:(1) Methodological Concerns on detection and quantification limitsSome methodological inconsistencies and limitations were observed that merit consideration. In Figure 1, there is a clear lack of consistency with theoretical expectations and with the trends observed in Figure 4A. Based on approximate calculations, having 10^-7 parasite equivalents with 100,000 target copies per parasite implies an average of 0.01 target copies per reaction. This would suggest an amplification rate of approximately 1 in 100 reactions, yet the observed 30% amplification appears disproportionately high. In addition, Figure 4A (not fragmented) shows lower values of positivity than Figure 1 for 10^-5 and 10^-6 dilutions showing this inconsistency among experiments. Some possible explanations could account for this inconsistency: (1) an inaccurate quantification of the starting number of parasites used for serial dilutions, or (2) random contamination not detected by negative controls, potentially due to a low number of template molecules.Similarly, Figure 5B presents another inconsistency in theoretical expectations for amplification. The authors report detecting amplification in reactions containing 10^-9 parasites after DNA fragmentation. Based on the figure, at least 3 positives (as I can see because raw data is not available) out of 388 PCRs are observed at this dilution. Assuming 100,000 copies of satellite DNA per parasite, the probability of a single copy being present in a 10^-9 dilution is approximately 1/10,000. If we assume this as the probability of amplification of a PCR (an approximation), by using a simple binomial calculation, the probability of at least 3 positive reactions out of 388 is approximately 9.39 x 10^-6 (in ideal conditions, likely lower in real-world scenarios). This translates to a probability of about 1 in 100,000 to observe such frequency of positives, which is highly improbable and suggests either inaccuracies in the initial parasite quantification or issues with contamination. In addition, at 10^-6 PE/reactions (the proposed limit of quantification) it is observed that 40% of repetitions are amplified. The number of repetitions is not specified but probably more than 50 according to the graph. Such dilution implies 0.1 targets per reaction (assuming 100.000 copies divided by 10^6), which means a total of 5 target molecules to distribute among the reactions (0.1 targets multiplied by 50 reactions). It seems highly improbable that 40% of the reactions (20/50) would amplify under the described conditions. Even considering 200.000 target copies per parasite implies 0.2 targets per reaction and an average of 10 molecules to distribute among 50 reactions. The approximate probability of the observation of at least 20/50 positives can be calculated by determining the probability of a reaction to receive targets by assuming a random distribution of the targets among the tubes, p = 1 - (1 - 1/50)^10, and then by using a binomial distribution to determine the probability that at least 20 reactions receive at least one target copy. The probability of at least 20/50 positive reactions in a dilution of 10^-6 parasites (200.000 target copies per parasite) is 0.00028. Consequently, the observed result is highly unlikely.

We disagree with the reviewer on both of these points.

First, the mean (S.D.) Cq values of the 10-3 PE unfragmented dataset in Figure 1 (40 replicates) and Figure 4a (88 replicates) are nearly identical at 30.02 (0.5813) and 30.21 (1.071), respectively, demonstrating a highly accurate initial quantification of parasites to make these 2 separate dilution series (reviewer’s point 1.1). At this concentration of parasites in blood, and with unfragmented DNA, each aliquot for PCR has an equal chance of receiving some parasite DNA (hence all reactions are positive) and a reasonably good chance of receiving similar amounts of parasite DNA (the Cq values cluster with relatively low S.D.). However further dilutions from this parasite input result in some aliquots that receive no parasite DNA and a much wider variation in the amount of parasite DNA/aliquot in samples that are positive (Cq mean (SD) of 34.47 (2.732) for 10-4 in Figure 1). This result demonstrates that these dilution series do not follow binomial distribution as suggested by the reviewer. This is likely because each template for amplification is not independently distributed. Instead, they are known to be clustered (on individual chromosomes or chromosome fragments) in the DNA. Indeed, this observation of widely varying Cq values in dilutions below 10-3 strongly suggested this clustering and was the impetus for fragmenting the DNA (see manuscript line 209). The impact of declustering achieved by DNA fragmentation supports this conclusion when the DNA is fragmented, 100% of aliquots are positive at 10-4 PE, 10X less than in unfragmented samples, and the Cq values are tightly grouped (mean 33.47, S.D. 0.3358), indicating the unequal distribution of targets upon dilution, rather than counting, pipetting errors or contamination as responsible for the lack of a binomial distribution of targets with increasing dilution. Thus, when entities are clustered and can’t be fully declustered, a simple binomial (or Poisson) distribution of counts cannot be assumed in the serial dilutions. Clustering results in more complicated distribution patterns, and it becomes difficult to predict precisely how these clusters will distribute from one dilution to the next (and thus differences in proportions of positives in different dilution series, as observed herein).

This clustering and unequal distribution of amplification targets also addresses the reviewer’s second comment with respect to the unlikelihood of detecting at least one positive at a high dilution. If we accept the reviewer’s estimate of 100,000 copies of target per parasite, then at 10-4 PE/aliquot - a dilution at which all aliquots are PCR positive in the fragmented samples (Figures 4a and 5b) – each aliquot would be expected to have on average 10 target sequences and the chances of detecting at least one positive reaction from 400 aliquots would be respectively 98% for the 10-7 dilution, 33% for 10-8 and 4% for 10-9 PE per aliquot. These percentages would change (increase) with a higher copy number of targets per genome, and if the targets are still clustered to some degree (which we would expect they would be even in the fragmented DNA). Thus, the chances of detecting positive PCRs at 10-9 PE is low, but it is not “highly improbable”.

Taking the reviewer’s second example of the frequency of positive reactions at 10-6 PE and the assumption of 200,000 target copies per genome (referring to Fig 5B, we believe), the mean template copies per aliquot would be 0.2 at this dilution. Assuming a negative binomial distribution of the still clustered templates (although mechanically fragmented, it would be highly unlikely that they would be completely declustered), then the probability of an aliquot being positive at the 10-6 PE dilution would be 16.7%. Our results in Figure 4A (26%) and Figure 5B (37.5%) are slightly higher but not “highly unlikely” as suggested.

We do not know the target copy number in the parasites used to make these serial dilution profiles herein but that is certainly different from the copy number in the parasites infecting each of the hosts from which we have analyzed blood. Thus, we do not propose that this assay can quantify the absolute parasite burden in a host nor do we see a benefit in trying to do so (see paragraph beginning line 384). Such quantification requires assumptions about not only the target copy number in the parasites in a host, but also that fragmentation is 100% efficient, and particularly, that a single or multiple blood samples accurately reflects the whole host parasite burden (clearly shown not to be the case with the data from serial bleeds presented in Figures 3 and 5). But we standby the conclusion that deep-sampling PCR when employed as presented herein, gives an accurate assessment of the presence of infection and relative parasite burden differences between hosts, and in the same hosts over time or under treatment and that the results presented are not compromised by inaccuracies in quantifying parasites for spiked samples or by sample contamination.

(2) Lack of details on contamination detectionAdditionally, the manuscript does not provide enough details on how cross-contamination was detected or managed. It is unclear how the negative controls (NTCs) and no-template controls were distributed across plates, in terms of both quantity and placement. This omission is critical, as the low detection thresholds targeted in this study increase the risk of false positives by contamination. To ensure reliability and reproducibility, future uses of the technique would benefit from more standardized and clearly documented protocols for control placement and handling.

We present a section in the Materials and Methods on preventing contamination and a case example when these precautions failed when preparing the dilution standards containing very high numbers of parasites. Directly responding to the reviewer, sixteen no template controls were included in every 384 well assay plate and we never obtained amplification products from those reactions. Additionally, as noted in the manuscript, uninfected macaques were negative on a collective >15,000 PCR reactions.

We understand the concern about contamination but we believe that we have taken the appropriate precautions and our data fully support that the positives we detect are real positives, not contaminations. It would be reckless to depend on a single positive PCR reaction out of hundreds to conclude that a host is infected; multiple samples must be obtained and analyzed to be certain in such cases, as we show exhaustively with the NHP samples here.

Rather than adding additional technical protocols such as plate layouts to this manuscript, we believe publishing a STAR Protocol or a similar detailed, step-by-step method paper would be more useful and that is our plan.

(3) Unclear relevance for treatment monitoring in HumansIn Figure 7A, the results suggest that the deep-sampling PCR method does not provide a clearly significant improvement over conventional qPCR in humans. Of the 9 samples tested, 6 (56%) were consistently amplified in all or nearly all reactions, indicating these samples could also be reliably detected with standard PCR protocols. Two additional samples were detected only with the deep-sampling approach, increasing sensitivity to 78%; however, these detections might be attributable to random chance given the limited sample size. While the authors acknowledge the small sample size in the discussion, they do not address the fact that a similar increase in sensitivity was reported in citation 5, where only 3 samples were tested with 3 replicates each. This raises an important question: how many PCR reactions are needed in human samples to reach a plateau in detection rates? This issue should be further discussed to contextualize the results and their implications.

We disagree with the reviewer’s conclusion here. First, it is not known how the “conventional” PCR would have performed in the human samples used herein as this was not done. However, it is very likely that it would have performed significantly worse for the following reasons. “Conventional” PCR for T. cruzi has a number of variations, but the most common approach is to mix whole blood 1:1 with a guanidine:EDTA solution, and then extract DNA for PCR from 100-300 ul of this mix. Thus, at best, one has the equivalent of 150 ul of blood that is being analyzed for the presence of T. cruzi DNA. In contrast, in the protocol described herein, we extract DNA from ~5 ml of blood and use aliquots from that DNA for PCR. Thus, even before fragmenting or deep-sampling, the approach described herein is sampling 33X more blood that the conventional protocol, thus likely increasing by over 30-fold the chances of detecting parasite DNA in blood from an infected subject. The smaller the volume of blood sampled as well as the number of samples obtained greatly impact the ability to detect T. cruzi infection in some hosts. This is clearly demonstrated in the extensive screening done in NHPs in this study and there is no reason to believe that the situation will be different in humans and dogs. So the relevance of these enhancements are clear for any host with T. cruzi infection; humans are not unique in this regard.

We don’t believe there will be a “plateau in detection rates”; individuals are either infected or not and the ability to detect that infection (whether with T. cruzi or any other pathogen) depends on the sensitivity of the test and the quantity of the sample available to be screened. Perhaps what is being asked is ‘how many PCR reactions have to be performed to be sure that someone is NOT infected?’. There is not a discrete answer to this and related questions, but by making some assumptions, one can make some estimates. The approach described herein is approaching single copy target detection and if this is true then one would need to PCR amplify ALL of the DNA from a blood sample to assure detection of that single template copy (so for a 200ug of DNA one might obtain from 5-10 ml of blood, 1600 PCR reactions of 125 ng each; 95% and 99% confidence could be obtained with 1520 and 1584 PCRs, respectively). But any conclusion from this testing applies only to that individual blood sample and we show clearly in the NHP studies that multiple samples have to be analyzed to detect parasite DNA in hosts with very low parasite burden – some samples contain parasite DNA and others do not. Thus hundreds of negative PCRs from a single or even multiple samples is unfortunately not definitive.

Such limitations exist for detection of any pathogen. A more important question for the future may be ‘is there a level of infection below which the risk of disease development is sufficiently low as to not be of concern clinically?’. Such is the standard in drug-controlled HIV infections, for example. The improvements we document in this work provides the means to answer such questions and additional improvements may be possible as well. But to be absolutely certain that a host is not infected by *T. cruzi*, one would have to sample some subjects (likely a small minority of the entire pool) multiple times and perform 1000’s of PCR reactions – as we done for the most difficult to detect macaques in this study.

Despite these limitations, this work represents a promising step forward in the development of highly sensitive diagnostic tools for *T. cruzi*. It offers a novel foundation for advancing the detection and monitoring of parasitemia, which could significantly benefit Chagas disease research community and clinicians focused on neglected tropical diseases. While addressing the methodological inconsistencies and improving robustness will be critical, this study provides valuable insights and data that could lead to future innovations in parasitological research and diagnostics.

As discussed in detail above, we do not agree that this study has any methodological inconsistencies nor that it lacks robustness.